# Experiential Learning: Conferences as a Tool to Develop Students' Understanding of Community-Engaged Research

Maria Zaharatos, Carolyn Taylor Meyer *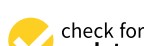 and Julian Hernandez-Webster

Middlebury Institute of International Studies, Graduate School of Middlebury College, Monterey, CA 93940, USA; mzaharatos@middlebury.edu (M.Z.); jhernandezwebster@middlebury.edu (J.H.-W.)
* Correspondence: cmeyer@middlebury.edu

**Abstract:** The purpose of this paper of practice is to explore the use of a "Conference as Curriculum" model to develop student understanding of critical approaches and challenges and opportunities in the field of community-engaged research (CER). Two higher education institutions in California's Monterey County sent 22 students to the "All-In: Co-Creating Knowledge for Justice" Conference in Santa Cruz, California USA in October 2022. The undergraduate and graduate students were funded through their academic institutions and accompanied by faculty and staff working on community-engaged research. Participation in the conference involved a pre- and post-conference convening to prepare students for the conference and then guide students through reflections on their learning and future work. The experiential learning activities offered in conjunction with the students' conference attendance were designed to: (1) foster students' connection to the community and each other; (2) develop students' understanding of community-engaged research; and (3) build students' professional acumen through attending a professional conference. Two authors of this article share their experience as student attendees at the conference. The student vignettes provide insight on the authors' learning experiences and offer design implications for the use of cohort conference attendance as an experiential learning activity. The article presents a conference experiential learning model that could be replicated and modified by other higher education institutions. We also place the project within a greater inter-institutional initiative to build a model for community-driven collaborations that seeks to address challenges surrounding higher education engagement with local nonprofits and governments.

**Keywords:** conference as curriculum; experiential learning; community-engaged research; university–community collaborations; critical participatory action research; higher education institutional collaboration

## 1. Introduction

Experiential learning approaches in higher education institutions in the United States (U.S.) have increased over the last 40 years (Buzzelli and Asafo-Adjei 2023; Kuh 2008; Selingo and Horn 2023). While some fields such as medicine and engineering have long touted the benefits of applied learning through co-ops and residencies, it is only in recent decades that there has been a push for nearly all degrees and disciplines to include experiential learning requirements such as internships, practicum, and field research, as documented in the US National Survey of Student Engagement (NSSE 2020; Stowe et al. 2022). Higher education institutions at the undergraduate level are promoting high impact (active learning) practices to boost completion rates, particularly among underserved student populations and first-generation scholars (Kuh et al. 2017; McDaniel and Van Jura 2022). While experiential learning and high impact practices have grown in their use in higher education institutions in the U.S., there remains limited literature on the use of professional conferences as structured experiential learning activities or high-impact practices, particularly within professional graduate degree programs. As higher education

grapples with heightened concern over the purpose and career value of a 4-year undergraduate degree (let alone a 2-year professional graduate degree), many thought leaders are promoting experiential learning activities such as project-based learning, micro-internships, and other resume-building projects (Selingo and Horn 2023). We argue that an increase in experiential learning projects and activities within the local community should be undertaken with and for the community as much as for the student's career development. We present Gordon da Cruz's (2017) critical participatory action framework as an approach higher education can employ to support just and meaningful engagements with community partners. We also build on Campbell et al.'s (2021) Conference as Curriculum model to promote the development of structured experiential learning experiences involving student attendance at professional conferences. There is a dearth of literature on the learning experience that conferences can provide post-secondary students. For example, Kuh's (2008) high-impact practices (HIPs) describe culminating experiences, but they do not detail conference participation as a high-impact practice. Koolage and Clevenger (2018) argue that HIPs should include conference learning experiences; however, their research is limited to a US undergraduate research conference hosted at their university. Furthermore, the Koolage and Clevenger (2018) model does not involve students at a professional graduate school (as we will discuss), and their research does not focus on developing students' awareness, skills, and networks to address social justice issues (although a goal of achieving gender parity in the field of study is proposed). Our research starts to fill this gap by presenting an experiential learning model to support a cohort of students attending a community-engaged research conference in a nearby city. Student participants in the experiential learning activity were also co-authors of this article. The student co-authors bring unique perspectives as participants in the activity. While higher education research may focus on the student experience, the student voice is rarely included at the author level in academic research. We include students as authors of this article to show the value of student engagement in the curriculum design process and as contributors to academic research on experiential learning.

This research also describes how two higher education institutions in the same California county sought to improve their approach to addressing inequities in community projects through a collaborative initiative. We describe how the CoLab initiative supported faculty, staff, and student participation at the "All-In: Co-creating Knowledge for Justice" (All-In Conference) in Santa Cruz, California in October 2022. The article provides an overview of CoLab's model and process for implementing a project that utilized a professional conference as a learning opportunity for 22 students, 4 faculty, and 1 higher education staff member across two California higher education institutions. We also discuss how CoLab's planning team explored approaches to community-engaged research (CER) at an institutional level, including applying a critical lens throughout CoLab's design and implementation process. The article includes vignettes from two co-authors on their experience as student participants in a Conference as Curriculum experiential learning activity. We present a model for how higher education institutions and experiential learning practitioners can build learning experiences around participation in professional conferences. This experiential learning model demonstrates how a Conference as Curriculum activity in the social justice space can expose students to critical community-engaged research processes and career paths.

### 1.1. CoLab's Story

The Middlebury Institute of International Studies (MIIS) and California State University Monterey Bay (CSUMB) developed CoLab in 2016 with the goal of making a tangible difference in addressing the complex "wicked" problems confronting our local communities in Monterey County[1]. CoLab's planning team, made up of faculty and staff, expressed concern that our community-universsity engagements were not leading to meaningful social change. Our concerns were not unique to our institutions, as higher education faculty and institutions have long grappled with how to develop more socially impactful

service-learning engagements (Marullo and Edwards 2000; Rosner-Salazar 2003). Both our institutions have rich local networks and a commitment to community-based learning. At CSUMB, over 3000 undergraduates complete degree-required service-learning hours with regional schools, nonprofits, and public organizations each year, resulting in over 90,000 h of annual service to the local community. Likewise, graduate students at MIIS complete organizational assessments, evaluations, and other strategic projects for local organizations as part of their graduate coursework.

Through critical self-assessment and conversations with our community partners in the years leading up to the development of the CoLab, both institutions realized the limitations of our approach. Community partners expressed concern over the lack of continuity between student projects and the limitations posed by trying to complete a project within the school's semester timeline. As such, CoLab was partially developed to address the continuity concerns created by the transitory nature of students who graduate and leave the community. To address this concern, the faculty/staff serve as stewards of the relationships with community partners so that the burden of transition is not borne by the nonprofit organizations. The extant service-learning literature explores the burdens, privileges, costs, and benefits of service-learning partnerships between the student and community (Blouin and Perry 2009; Cronley et al. 2015). Callan (2020) argues that service-learning research focuses on how universities and students benefit from service-learning with no consistent empirical evidence showing how service-learning benefits community partners. Furthermore, Driscoll (2014) argues that the Carnegie Classification of Community Engagement study shows more institutions focusing on student benefits than impacts on the community. Lack of student commitment and academic calendar limitations are cited in the literature as negatively impacting the community organization and the university–community partnership (Blouin and Perry 2009; Sandy and Holland 2006). Most higher education institutions do not explore the long-term impact of their service-learning work with community partners, or the effects institutions have on addressing the root causes of issues significant to the community (Callan 2020). CoLab draws on critical participatory action research practices to emphasize the practice of regularly reviewing and questioning the power and resource distribution in university and community partnerships. The privilege higher education institutions hold in knowledge making on and for the community is documented in research on service-learning and university–community partnerships (Preece 2016; Fisher et al. 2004; Bortolin 2011). We wanted CoLab to dismantle the idea that universities are the purveyors of knowledge and that the research produced through higher education institutions was of greater value than the research produced by community members.

Based on this assessment, the first iteration of CoLab was developed to build meaningful relationships with community partners that would address the continuity and communication issues shared by our community partners. We sought to transform the community–university relationship into a partnership focused on the co-creation of knowledge and the sharing of resources and skills (Guni Network 2013). CoLab was created with the following goals: (1) commit to longer-term, strategic partnerships; (2) alter our stance towards the knowledge and insight that resides within communities and community members; (3) rethink our assumptions about rigorous research; (4) expand our vision of expertise (beyond faculty); (5) make impact on social, economic, and political issues; and (6) forge new impact-oriented collaborative mechanisms (Note 1).

The development of CoLab led to the sharing of resources between the two higher education institutions, including collaborative lists of community projects completed by undergraduates at CSUMB and graduate students at MIIS. During the first two years, we held convenings with our faculty to foster a community of practice around community-engaged research (CER) and separately with our community partners to learn more about their experiences engaging with higher education institutions. Such discussions led to deeper engagement with one local city government in Monterey County and resulted in multiple projects completed by CSUMB and MIIS at the request of the local city partner. Unfortunately, other issues took priority at the start of the global pandemic in 2020 and this

initiative lost traction. Changes in staffing at the partner city government also complicated the relationships we hoped to foster.

Since the start of the pandemic, CoLab has increasingly focused on building the capacity of our structure so that we might be better able to support our community partners. In 2022, CoLab started to develop into a Community of Practice (CoP) model for faculty, staff, and students working on critical CER. Our model was influenced by research on the benefits of CoPs in building good practices in higher education settings (Sánchez-Cardona et al. 2012) and the questioning embedded in critical community-engaged scholarship (Gordon da Cruz 2017). The model's philosophical underpinnings also draw on relational approaches to aspirational justice-oriented goals (Avineri and Martinez 2021). Positioning CoLab as a CoP model that shares CER resources and promotes students, faculty, and staff to be engaged in critical, self-reflective, community-based research became the most feasible starting point for an initiative that lacked funding and a formal institutional home. This approach was central to the focus on sponsoring students, faculty, and staff to attend the All-In Conference in October 2022. MIIS students attending the conference were exposed to CER and participatory action research (PAR) practices that they had not yet encountered in their graduate coursework. The CSUMB students had been exposed to and engaged with PAR and CER prior to the conference. The conference sessions covered content that could not have easily been offered through an on-campus training opportunity at MIIS due to staffing and funding constraints. It is important to note that the CSUMB educational system provides more PAR and CER opportunities due to the local mission of the university, while MIIS is a much smaller educational institution (less than 1000 students) that has historically focused on international work and research.

*1.2. Our Approach*

As with many higher education institutions, the philosophical underpinnings for both CoLab and our approach to preparing and supporting students to attend the All-In Conference can be traced to the research areas of CoLab's planning team and the educational tracks at our institutions. Both CSUMB and MIIS prioritize values and skills associated with intercultural competence and interculturality. MIIS' emphasis on intercultural competence, for example, explores student self-development through behavioral skills (self-awareness) and cognitive abilities (processing and situating new information). CoLab also utilized the teaching and research experience of one faculty team member to implement a critical service-learning framework, which places more attention on social justice goals, power dynamics, and authentic relationships than traditional service-learning models (Mitchell 2008).

Additionally, several team members have deep connections in participatory action research. Our team members value the use of PAR for its focus on democratic research that is conducted "with participants" rather than "on participants" (Kindon et al. 2007; Wood 2019). PAR's asset-based approach dovetails with CoLab's use of appreciative inquiry to draw on community members' knowledge and expertise in promoting positive social change. Our institutions and planning team support the use of restorative justice practices to foster environments where communities and organizations can address deep-seated beliefs and effectively process conflict or harmful events (Costello et al. 2019). CoLab was also influenced by transformative justice approaches that recognize larger systemic issues such as socio-political and economic challenges (Coker 2002). Both of our higher education institutions have increased training and awareness around diversity, equity, inclusion, and justice like many higher education institutions in the US over the last five years (Abrams 2022). Our planning team grapples with the "push and pull" of focusing on the personal development of our learners (e.g., intercultural competence and restorative justice) without losing sight of the larger systemic issues embedded in diversity, equity, and inclusion discussions. We seek to find the appropriate balance of different levels of analysis and the most effective approaches to systemic change in nearly all our CoLab planning meetings. The aforementioned approaches feed the emerging critical CER practices we hoped to cultivate through CoLab.

Both institutions support service-learning and experiential learning approaches through courses and degree requirements. Our planning team includes faculty and staff tasked with building and supporting applied learning opportunities that connect students with the real world and their desired professional fields of practice. As previously described, CoLab has sought to institutionalize meaningful community engagement and develop capacity for community-engaged scholarship at CSUMB and MIIS. We seek to cultivate community-engaged scholarship that emphasizes critical race theory as described by Gordon da Cruz (2017). Gordon da Cruz (2017) describes community-engaged scholarship (CES) between universities and communities as including the following components:

(1)    *"Real-life social problems in our democracy are defined with or by the community*
(2)    *These real-life social problems are investigated in scholarly ways*
(3)    *Community–university partnerships are collaborative and mutually beneficial*
(4)    *Knowledge to address public issues is collaboratively developed with community and university members*
(5)    *Institutional resources are utilized to address these real-life public issues*
(6)    *Community research or projects are integrated with faculty members' research and teaching"*

Gordon da Cruz (2017) draws on critical race theory approaches to develop a critical community-engaged research model with the following additional questions for researchers to consider when working with community partners:

1.    *"Are we collaboratively developing critically conscious knowledge?*
2.    *Are we authentically locating expertise?*
3.    *Are we conducting race-conscious (instead of color-blind) research and scholarship?*
4.    *Is our work grounded in asset-based understandings of community?"*

CoLab faculty, staff, and student leaders pose this line of CRT questioning as we grapple with the mission and approaches to our work with community partners on pressing local issues.

## 2. Interventions & Implementation

### 2.1. Developing Community-Engaged Researchers

The inspiration for promoting student attendance and participation in the All-in Conference stemmed from earlier research at MIIS on the model of "Conference as Curriculum" (Campbell et al. 2021). In a 2019 study of 13 MIIS graduate students who attended an academic or professional conference on international education, the researchers presented an experiential learning model that used a professional conference as a learning opportunity. The authors found that students gained insight into professional and scholarly trends, explored the diversity of the conference body, and positioned themselves within their professional community (Campbell et al. 2021). These findings inspired CoLab to embark on a similar project of engaging student participation in the All-In Conference. The Conference as Curriculum activity was designed as a training opportunity to better prepare those students who may go on to complete project work with local community partners either in the local county or in other localities around the country and world. We sought to develop the community-engaged research capacity and understanding of our students, faculty, and staff through conference attendance.

As previously described, CoLab was developed to institutionalize meaningful community engagement and develop capacity for community-engaged scholarship at CSUMB and MIIS. The CRT questions posed by Gordon da Cruz (2017) are central questions asked by CoLab faculty, staff, and student leaders as we grapple with the mission and approaches to our work alongside the community on pressing local issues. While some of the classes available at the two institutions take students out of the traditional classroom, the "Conference as Curriculum" framework adopted by CoLab sought to create an extra-curricular CER training opportunity that would expose students to a broader regional and national network of CER practitioners, funders, and partnerships. The conference used in this intervention shared many resources and experiences relating to the aforementioned critical

CES components. The conference's focus on justice and critical race theory was of interest to CoLab and its scholars to support our efforts, " . . . in producing knowledge that more effectively dismantles systemic sources of racial and social injustice" (Gordon da Cruz 2017, p. 363).

### 2.2. Technical Implementation Considerations

Figure 1 summarizes the five phases of the design and implementation process detailed in the following sections. It outlines major decision points and activities in the design and implementation of the project at the pre-, during, and post-stages of the Conference. The project took place from August to November 2022 and was managed by a CoLab Graduate Assistant/student worker (GA). The GA was charged with managing the project, providing logistical support, and serving as a coordinator between student participants and CoLab's planning team. The GA served as an organizer and facilitator of cohort activities and events throughout the conference phases and provided personalized logistical support for students upon request. Having a designated management role ensured timely completion of project objectives, relieving the planning team faculty and staff of administrative and project-related burdens.

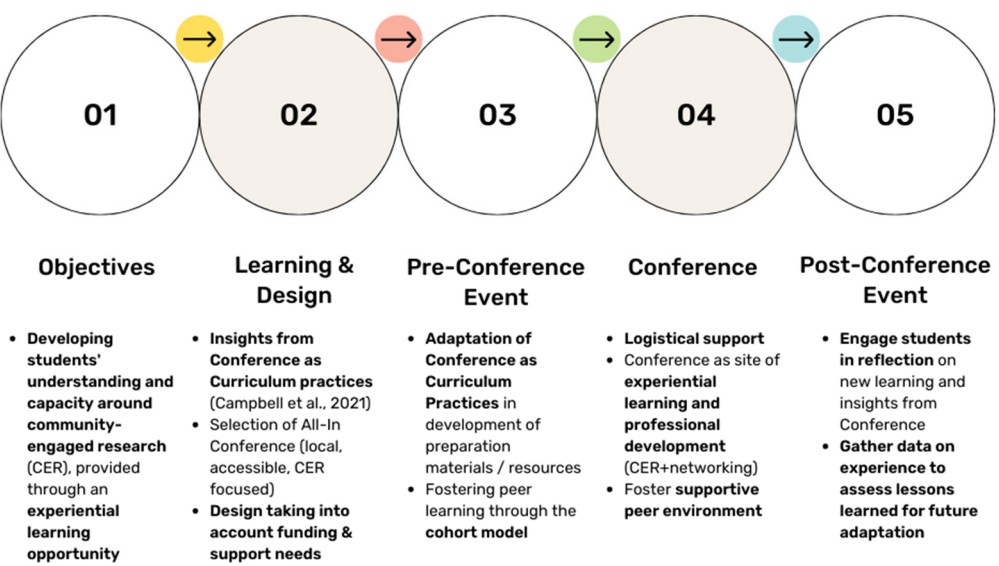

**Figure 1.** Intervention Process. This figure captures the five main steps of an experiential learning intervention that took place over four months in 2022 (Campbell et al. 2021).

### 2.3. Intervention Design Considerations

The conference was selected for its alignment with the Initiative's goals and potential to build student and faculty capacity for CER. While the event was national in scope, the conference brought together numerous local and regional organizations, allowing CoLab and the student cohort to gain exposure to local community issues such as housing insecurity, youth empowerment, and public health challenges.

Located in the neighboring Santa Cruz County, the conference was within commuting distance for attendees from both institutions, which also permitted greater accessibility. Conferences can be considered an investment of time and money in one's career and network (Cherrstrom 2012) and supportive structures such as funding significantly impact many students' ability to engage in these types of extracurricular opportunities (Omolabake et al. 2019). All conference registration costs were covered by CoLab. Graduate students were also eligible to receive $250 stipends that were prorated depending on if the students could attend 1 day ($75) or 2–3 days ($250). Students decided the time commitment that best fit the demands of their course and work schedules, having the option of attending just one day or two to three days of the three-day conference. This built-in flexibility allowed for maximization of available funding resources and increased the number of students who

could receive financial support. Students also had agency in decision-making with respect to the funding they received, allowing the students to determine how to spend the stipend most appropriately in offsetting their lodging, transportation, and other costs.

The conference was located approximately one and a half hour's drive from the students' campuses. The cost to send 18 graduate students to the conference, pay for graduate assistant time to coordinate the cohort meetings, and cover the cost of supplies and two group lunches was less than $7000 USD in 2022. CSUMB undergraduate students were supported separately through their campus research center. Full-time faculty and staff time were not included in this cost, as their contributions were considered part of their regular employment with the higher education institutions. The funding was allocated for this specific activity due to: (1) the conference focus on community-engaged research, (2) the ability to use a cohort model approach, and (3) the proximity of the conference to the higher education institutions' location. The funding for this experiential learning initiative was allocated through the graduate school's provost office.

Similarly to the process of graduate and post-graduate student socialization itself, in which inequalities exist between students' social capital regarding scholarly norms (Gopaul 2016), not all students at the two higher education institutions had the same social and financial capital. For example, one of this article's student co-authors shared this experience:

> [W]hen school started up again in the fall, I was forced to drop my restaurant job to continue my internship through December. With no income and no shortage of expenses, I knew I could only spend on essentials even if it meant missing out on impactful opportunities or exciting outings. As a result, I initially declined the chance to participate in the All-In Conference,(despite its relevance and alignment with my coursework and professional interests) until I was informed that funding was available.

CoLab is deeply rooted in social justice and equity, which made it important to provide a financially supportive environment for students and help ensure student access to the learning opportunity.

*2.4. Pre-Conference Preparation Event*

Preparing for a conference can: (1) allow attendees to familiarize themselves with the schedule so as to be focused on discovery and networking during the actual event; and (2) help optimize the learning opportunity by framing important questions and practices for professional and academic development (Cherrstrom 2012).

In preparing students for the conference, CoLab relied on the "Conference as Curriculum" model and other relevant literature (Campbell et al. 2021; Cherrstrom 2012). First, students and faculty were primed on the subject of community-engaged research (CER) through a series of readings shared by email and digital folder access beforehand: a 1-page description and white paper about Colab; "Community-Engaged Research: A Quick-Start Guide for Researchers" (Handley et al. 2010); and "Chapter 2: Knowledge, Higher Education and the Institutionalization of Community-University Research Partnerships" from Strengthening Community University Research Partnerships: Global Perspectives (Hall et al. 2015).

The following table (Table 1) summarizes the adaptations to the "Conference as Curriculum" model (Campbell et al. 2021) made by CoLab for this intervention. It highlights differences and potential enhancements that build off the original model, including self-guided resources, a strong cohort model, and increased accessibility of the learning opportunity through financial support.

**Table 1.** Building off the "Conference as Curriculum" model.

| | Elements Adopted and Adapted from Campbell et al. (2021) | New Elements in Our Initiative's Model |
|---|---|---|
| Pre-Conference | • Prepare: set goals and make a plan<br>• Gather information (logistics, conference, speakers etc.) | • Goal-setting framework<br>• More explicit/reinforced cohort structure |
| During Conference | • Experience and attend sessions attentively to capture emerging trends from the field<br>• Ask and engage with critical questions<br>• Connect authentically with others + prior knowledge<br>• Reflect | • Optional cohort dinner<br>• Increased coverage of financial costs (registration+stipend) |
| Post-Conference | • Discuss and share new learning and perspectives<br>• Reflect on the learning process<br>• Explore new opportunities and networks | • Provide resources/connection to related opportunities through email updates |

Note. Table shows modifications to the Conference as Curriculum model (Campbell et al. 2021) implemented through this Initiative.

Additionally, in advance of the pre-conference convening, preparation materials were created for students to enrich and turn the conference into a learning experience. The GA, utilizing practices from the "Conference as Curriculum" model and effective conference practices (Campbell et al. 2021; Cherrstrom 2012), developed an original framework/worksheet to invite students to link their curricular learning and projects with specific sessions and speakers. This Goal-Setting Framework (Figure S1) was built on the aforementioned best practices by enhancing preparation/orientation for learning and networking with specific sessions and presenters. For example, the format allowed students to link their curricular learning and projects with specific sessions and speakers before the conference began. Additionally, the inclusion of an "Alternative" category provided students with room to explore options for when their interests change, and they may seek to deviate from their proposed conference plan (Cherrstrom 2012). Students were encouraged to utilize these self-guided resources before, during, and after the conference.

At the pre-conference event, the GA introduced the self-guided learning materials and, in small groups, students shared their existing knowledge and experiences with CER and community engagement more broadly through past professional experiences, service-learning, etc. This pre-conference event framed and provided "structured support" for the conference as a learning conference and site of participatory action learning (Wood et al. 2017, p. 122). As such, students could draw connections between the learning that happens at conferences and their curricular learning and specific projects and goals.

Furthermore, as included in the original "Conference as Curriculum" model (Campbell et al. 2021), we sought to develop a community of practice model to foster peer support and co-learning across students from both higher education institutions. By taking part in a professional learning community (or cohort) with peers and faculty guides, students could share their learning with other students going through a similar experience. Students were first introduced to one another at the in-person Pre-Conference Preparation event, where they shared lunch and fostered initial (or continued) relationships in the MIIS community garden.

Finally, beyond simply meeting together, we created online platforms for students to connect, including: (a) a shared student biographies resource; (b) a WhatsApp group chat; (c) a shared spreadsheet to help coordinate ride/hotel sharing; and (d) a collective note space for students to share their learning and takeaway with others. These digital tools were used not only to facilitate relationship-building but also to provide opportunities for solidarity and peer support for students. One student co-author reported:

> *I think the in-depth preparation for the conference and having a friendly support system and community of practice through the cohort is what made the experience so deeply*

*rewarding and what encouraged me to feel comfortable making connections, networking, and asking questions alongside the researchers, practitioners and activists present.*

### 2.5. Conference

Seventeen students and three faculty and staff from MIIS as well as five students and two faculty from CSUMB attended the All-In Conference, engaging in presentations in plenary and breakout sessions, networking events, and cultural and poetic performances. An informal cohort dinner was organized by the GA at the midpoint of the conference with the goal of offering a safe space for reflection. Indeed, using a cohort structure can mitigate fears students may have about their interactions with the other conference participants and help instill a sense of power and opportunity in the learning experience (Hyland and Kranzow 2012). A student co-author of this article described the benefits they experienced through the cohort model as someone who had little experience at professional conferences, particularly an in-person conference:

*I worried if I had the right clothes, enough knowledge, or the appropriate cultural competence to avoid sticking out from the crowd as someone who might not quite belong there. [. . . ] Within our group, I quickly built connections and relationships, and learned that some of my concerns were shared among others. I benefited from the experience and advice of my colleagues with more familiarity in conference environments, who guided me in managing my expectations, as well as the social support of friends in the group who facilitated and elevated my participation through their encouragement.*

During the conference, CSUMB students led panels on their service-learning work and CoLab's planning team hosted a roundtable session on their inter-institutional university-community partnership model. The CoLab roundtable session included a conversation with attendees about CoLab's relational and aspirational goals for community partnerships in Monterey County (Avineri and Martinez 2021; Hall et al. 2015). The panel also provided attendees with the opportunity to reflect on lessons learned from the conference.

### 2.6. Post-Conference Debrief Event

Lastly, similar to experiential learning, literature on learning at conferences emphasizes the importance of reflection as a key aspect of learning, which informed our framework for planning a Post-Conference Debrief session (Campbell et al. 2021; Kneale et al. 2016; Kolb 2014). In their 2017 paper, Wood et al. offered the conclusion that "all conferences can be made into learning conferences through a support program consisting of pre- and post-conference workshops using [Participatory Action Learning and Action Research (PALAR)] principles [. . . ] to make them a space for maximizing postgraduate learning and development" (p. 133). Thus, the conference cohort and CoLab's planning team met a week later at the CSUMB campus garden to debrief on their learning as a group, reflect on challenges and emerging practices, as well as share future community-engaged opportunities. For those who were able to attend, both the pre- and post-conference convenings allowed for participants to mingle, share their interests, and build relationships outside of the conference in a non-traditional learning setting.

### 3. Learning from the Project

### 3.1. Methodology

This paper is first and foremost a paper of practice, presenting a model for university–community collaboration and Conference as Curriculum that is still a work in progress. Earlier in this article, we presented the history and development of CoLab, a cross-institutional higher education collaboration focused on improving university–community collaboration and CER practices. The following section presents exploratory research in the form of vignettes from two co-authors of this article who were also student participants in the Conference as Curriculum experiential learning activity described in this article. In an effort to remove the higher education institution from dictating the parameters of the exploratory qualitative research, the student co-authors developed the research question and their

responses. The student co-authors developed and answered the research questions "What are my perceptions of the impact the conference had on my learning, particularly around community-engaged research?" and "How would I describe my experience engaging with the Conference as Curriculum model?" The student attendees presented their perception of learning and impact without going through the sieve of a third researcher. As the researcher and participant, the student co-authors controlled what they chose to share. We recognize that this model does not fit squarely into an existing qualitative approach. Nevertheless, as proponents of critical participatory action research, we felt the student voices should be shared and controlled by the student co-authors. We share these exploratory reflections with the goal of encouraging empirical research of the model and approach.

Thus, this paper of practice features vignettes from the two students, who also served as co-authors. One student co-author co-designed and led the Conference as Curriculum pre- and post-conference briefings. The other student was invited by the other co-authors to co-author this article. The inclusion and leadership of the students who participated in the learning intervention as authors of this article aligns with the participatory approaches valued by CoLab and those that the conference learning experience hopes to foster. It was essential to the ethos of our project to include some of the perspectives and voices of stakeholders, in this case students, who stand to benefit from the "Conference as Curriculum" experiential learning model. Our inspiration for this approach also stems from the words of Matika Wilbur, a visual storyteller from the Swinomish and Tulalip peoples of coastal Washington who describes an indigenous approach to storytelling that ensures the participant controls how their story is told (Wilbur 2023). Student-Author A provides a perspective as one of the student attendees and Student-Author B provides a dual-perspective as both a student attendee and the primary facilitator/implementer of the model.

The vignettes thus represent data for exploratory qualitative research, drawing on a participatory approach. They provide insight into the impacts of the conference learning experience on students and their understandings of CER practices and commitments to social justice. The vignettes were drafted by the student co-authors three months after the conference took place, with the authors documenting their perception of their own learning about community-engaged research for social justice at the conference and their experience with the Conference as Curriculum model. Writing the vignettes also served as a process of metacognition, as the student co-authors reflected on their learning and learning process as it related to their attendance at the conference and participation in the Conference as Curriculum experiential learning model. We practiced reflexivity in presenting the following vignettes through student co-author discussions and self-reflection that included no interference from the non-student co-author or other higher education stakeholders.

As authors of the article, the student co-authors were able to learn from and dissect their learning experience at the conference. An exploratory thematic analysis was conducted by the student co-authors to pull out emerging themes and insights, as unpacked in the following sections.

### 3.2. Student-Author Vignettes

Student-Author A

> I have always lived in circumstances that have required me to work while I am in school. So, when I moved to begin my graduate studies at MIIS, I also found work as a server at a local restaurant. I continued to work nights and weekend shifts through the spring and then over the summer to complement my internship, which was unpaid, in the absence of classes. However, when school started up again in the fall, I was forced to drop my restaurant job to continue my internship through December. With no income and no shortage of expenses, I knew I could only spend on essentials even if it meant missing out on impactful opportunities or exciting outings. As a result, I initially declined the chance to participate in the All-In Conference, despite its relevance and alignment with my coursework and professional interests, until I was informed that funding was available.

*With this financial assistance, I was able to take advantage of the opportunity and further invest in my learning, professional network, and personal growth.*

*This was the first academic or professional conference that I have attended, and in the weeks leading up to the event I was growing increasingly nervous. I worried if I had the right clothes, enough knowledge, or the appropriate cultural competence to avoid sticking out from the crowd as someone who might not quite belong there. This is where the cohort became such an impactful feature of my experience. Within our group, I quickly built connections and relationships, and learned that some of my concerns were shared among others. I benefited from the experience and advice of my colleagues with more familiarity in conference environments, who guided me in managing my expectations, as well as the social support of friends in the group who facilitated and elevated my participation through their encouragement. Finally, cohort discussions of key themes and interesting new ideas were highly engaging, and often I found myself learning just as much from my peers as I did from the speakers throughout the conference.*

*The cohort and CoLab organizers also made the logistical pieces of the puzzle fit seamlessly with regards to factors of access such as transportation, hotel selection and reservations, and dining. By establishing communication lines within the group early, we were all able to organize shared rides to and from the Conference, split the cost of hotel rooms, and take advantage of each other's local knowledge when it came to food. I have no doubt that the cohort model contributed greatly to my experience at the Conference—I am grateful to all the other participants involved in the group, and I shudder at the thought of having attended as an individual knowing, now, the benefits of attending as a cohort.*

*From the food provided at pre- and post-conference meetings, which could only be held around lunch time due to scheduling needs of the group, to the financial and logistical support from the facilitating institutions, it is clear that CoLab is deeply committed to the principles of equity and student and engagement that we explored throughout the Conference. I did not really know what to expect going into this experience, but more than anything else I left feeling inspired and proud. I met so many brilliant minds and kind souls who are working tirelessly to make our communities safer, more prosperous, and more inclusive that I felt a surging hope for positive change. In sharing my feelings with the cohort, I learned that many of my peers felt it too. It was a special thing to explore concepts and practices learned through our coursework by engaging with professionals and academics in public, private, government, and university spaces.*

Student-Author B

*In addition to getting to attend the Conference as a student, I was responsible for assisting the CoLab team with the planning and execution of the conference attendance and preparation, as their current Graduate Assistant (GA). This GA role involved recruitment of participants, handling logistics, and developing and coordinating the programming for our pre- and post-conference events. During the conference itself, I also juggled attending sessions and serving as a resource for my peers, facilitating carpooling and coordinating a cohort dinner. Holding these numerous roles and positionalities, I felt like I was able to learn so much more from the experience, both building my knowledge and skills in community-engaged research but also in terms of facilitating learning and engagement for others and for myself. Writing this paper has also given me a more holistic understanding to reflect on how this conference learning experience impacted my growth relating to social justice practices and professional development.*

*Firstly, developing the CoLab events allowed me to take a deep dive into the field of educational and professional development, such as with the "Conference as Curriculum" framework. At MIIS, I am pursuing a Masters in Public Administration, hoping to work in social change and community development, with a focus on marginalized youth. The conference was truly inspirational and I felt grateful to be exposed to and engage in dialogue with so many amazing people, projects, and organizations from across the country. For me, the conference reinforced my belief that true social justice work begins at*

*the local level and engages in equity and authenticity. I learned key skills and insights on everything from allyship to high-leverage partnerships. It also raised various questions for me about my career, in the sense that I am still exploring whether I want to focus my work domestically or internationally; I was drawn to many of the attendees' commitment and connection to their communities and have since been in the process of grappling with where and how I define my sense of community and place.*

*Ultimately, like many other students, I would not have been able or had such a strong incentive to attend had it not been for the funding that CoLab provided. Additionally, this experience was initially new and daunting, as I only had experience with attending an online undergraduate research conference in the past (due to the COVID-19 pandemic). I think the in-depth preparation for the conference and having a friendly support system and community of practice through the cohort is what made the experience so deeply rewarding and what encouraged me to feel comfortable making connections, networking, and asking questions alongside the researchers, practitioners and activists present. I am really inspired and excited to apply some of these new pieces of knowledge and connections in community-based work through upcoming projects and after I graduate!*

### *3.3. Reflection*

Our vignettes shed light on both student learning and key conditions for successful implementation of the model. We share common reflections on the importance of the opportunity being funded for students as a critical component for access. We also highlight the benefits of the cohort structure, including learning from and finding confidence and support in our peers in order to have a better overall experience and learn in a CoP. This was especially useful for us as first-time attendees of an in-person professional conference. Additionally, for graduate students in the cohort, like us, with limited or no prior experience with community-engaged research, the All-In Conference introduced basic CER skills and strategies.

The conference allowed us to learn about and take interest in CER and PAR. Participants at the Conference were exposed to examples of CER through presentations about various projects and university-community partnerships on social justice issues. The broad array of CER topics—everything from indigenous and Latinx empowerment to student and youth-led food sovereignty projects—meant that there were rich and diverse perspectives and participants for everyone to learn from. Though the many organizations and institutions represented at the conference shared common goals in pursuing social justice, their methods and stories were unique. These multi-disciplinary, multi-cultural and multi-sectoral perspectives exposed us to new strategies for community engagement and the importance of taking a grassroots approach that listens to and learns from community members while valuing community knowledge and expertise. The conference highlighted tools for and the importance of creating equitable partnerships including: centering community interests, inclusive stakeholder participation, and authentic relationship-building. This new learning also drew upon concepts and practices we have studied in our coursework, with the added benefit of exploring these topics through engagement with community activists and professionals across academic, public, and private sectors.

Engaging first-hand with the scholars and community-based organizations at the conference also brought new questions and ideas to the forefront, both about the field and personal identity and career goals. Most importantly, the conference bolstered our existing commitments to social justice work, demonstrated the value of critical community-university engagement, and created an opportunity to reflect on one's purpose and activism. Finally, our vignettes also serve to highlight a sense of inspiration and motivation to further get involved and a desire to share what we have learned.

### 4. Limitations

Our model and approach included logistical, financial, and design limitations. As with any conference, the timing of the event did not fit with every student's schedule. The

All-In Conference was held in late October, around the middle of both higher education institutions' fall semesters. Most of the MIIS participants had only just started their graduate degree eight weeks prior with little time to familiarize themselves with Monterey County and CoLab. Furthermore, coursework grounded in PAR and CER at MIIS primarily takes place during the spring semester. As the conference took place during the fall semester, there were no credit-bearing courses associated with the "Conference as Curriculum" opportunity for graduate student participants. Most CSUMB undergraduate participants in the conference were more familiar with the area and with PAR and CER approaches through their coursework. Several of the CSMUB students were presenters at the conference. No graduate student participants presented at the conference. In future models, we recommend designing a Conference as Curriculum course and exploring mentorship models for students, as developed by Flaherty et al. (2018).

We are not able to determine whether the experience would have offered a richer learning experience if attendance at the conference was part of a full-semester course. Despite offering funding to fully cover the cost of registration and offset the cost of travel and accommodations in the nearby city, the cost may have still been prohibitive for some students grappling with other costs and expenses associated with their education. The experiential learning opportunity was available to all students who expressed interest and signed up for the event by the deadline. The opportunity cost of not working during the days of the conference may have prevented some students from attending. A final logistical limitation was the heavy course load in higher education degree programs. The course load may include nearly 50 h of work a week, leaving little time for exploration of non-degree learning options that enrich the learning environment.

We recognize that providing students with the option to attend for one day instead of requiring attendance at all three days of the conference may have impacted learning outcomes. This question is beyond the scope of this article; however, we structured the experiential learning activity to be flexible for accessibility purposes. We designed the activity with the understanding that conference attendance for at least one day would accomplish the goal of exposing the students to a community of practice and community-engaged research approaches including the norms, concerns, and trends in the field. As many of our student participants were newcomers to community-engaged research practices, the ability to attend for one day may have given participants the initial exposure to a new field that may plant the seed for future learning and practice.

Financial sustainability is another limitation of the "Conference as Curriculum" model, as schools must devote financial resources for student travel and attendance at conferences. Higher education institutions must make funding cohort conference participation a priority within the annual budget. Additionally, conferences on this subject matter may not take place in our geographical area on an annual basis. For instance, the Conference is not an annual event in Santa Cruz, California. Replication of our 2022 model will be more difficult if a conference requires more expensive travel and a longer time commitment for students to travel to the conference's location. The value of a local conference included the ability for students to attend for only one day if their course and work schedules did not permit attendance at all three days of the conference. Our model did not include presentations by any of the graduate students, given the limited exposure of the graduate students to the content and local community at the time of the conference. Despite funding challenges, most higher education institutions offer student funding for conference participation; however, there is little use of the cohort model or integration of the Conference as Curriculum model as part of a university or college center initiative.

Our model did not include applied learning projects immediately after the conference; however, these types of projects are ongoing at CSUMB and graduate students interested in the content of the conference could take two courses at MIIS related to the conference themes during the subsequent spring semester. Our model was also limited by a lack of capacity to monitor the impact of the conference on the future careers and academic endeavors of students. While this paper of practice highlights the conceptualization and

implementation of this model, additional research and empirical studies are needed to show the impact on student learning beyond self-reported student-author vignettes. We acknowledge that these self-reported student vignettes are anecdotal, limited to the two student co-authors, and do not demonstrate evidence of learning or student development across the cohort. We also acknowledge that our study cannot fully address or prevent the chance of volunteer bias, as one of the student co-authors completed the research as part of a graduate assistant position and the other co-author was invited to co-author the article. We also acknowledged the potential for positivity bias in our author meetings. Student co-authors reflected on the potential for this bias to occur prior, during, and after drafting their responses.

## 5. Implications

By sharing this paper of practice on the use of a conference as an experiential learning opportunity, we hope to inspire other higher education institutions to integrate funding and curricular support for student attendance at conferences. Our model presents one approach to scaffolding the learning experience through pre- and post- conference events with learning guided by student leaders, faculty, and staff. The model shares tools students can use to self-direct their learning and prepare for a professional conference, whether it is their first or one of many. Attendance at the conference exposed students to contextual learning and first-hand challenges that community-engaged research professionals often experience. The experiential learning activity aimed to inspire students to seek careers and current and future work in community-engaged research and employ a critical community-engaged research approach to social justice efforts. While a semester-long course might bring in guest speakers every other week, the Conference as Curriculum model presented a flexible and time-efficient alternative option for connecting the professional world with the classroom that could be explored in future research.

CoLab's use of the conference experiential learning model and insights from student vignettes align with existing research framing conferences as sites of learning and socialization for scholars (Campbell et al. 2021; Chapman et al. 2009; Cherrstrom 2012; Kuzhabekova and Temerbayeva 2018; Omolabake et al. 2019) and contributes emerging support for the use of experiential learning, such as conferences, in CER training. The conference allowed for real-world engagement with CER practitioners and community-based organizations, providing students with the ability to contextualize their academic understandings of social justice work. Our student-author vignettes also suggest that opportunities like these may be even more valuable in today's hybridized (in-person and online) academic and professional landscape post-COVID, where student learning and socialization may be limited in a virtual conference, for example (Campbell et al. 2021).

Furthermore, the learning that students could take away from the conference was not only a series of steps or how-to's but nuanced frames with which to understand research and the complex ecosystems of partnerships. From the perspective of students' socialization, the vignettes support the idea that conferences allow students to develop their critical and independent thought, all the while engaging with a professional learning community of peers, practitioners, and academics (Hyland and Kranzow 2012). The student-authors suggest that "Conference as Curriculum" good practices include the value of a strong cohort structure as well as the importance of self-care and time for reflection, which supports some emerging findings on the model (Campbell et al. 2021). Our reflections point to the importance of conferences, beyond academic and professional growth or socialization, towards an activity that fosters personal growth, centers equity, and enhances relationships through peer learning in the cohort structure. The conference environment, such as the one provided by the All-In Conference, was generally characterized as inspiring, and, similarly to community-based PAR, can be a source of praxis driving the embodiment and reshaping of new justice-oriented ideas and collaborations (Fine and Torre 2019).

The social justice elements of this research include an overview of the efforts of two higher education institutions to build long-term meaningful engagements with local

partners in order to address social issues affecting the region. We describe how CoLab and other higher education institutions can employ Gordon da Cruz's (2017) community-engaged scholarship and critical race theory questions to their work with the community from the design stage forward. Traditional research agendas can maintain a top-down approach that positions the higher education researchers with more power and privilege than the community partner. Community-engaged research inherently focuses on the social justice issue of power distribution and the building of trust between researchers and the community, which often includes marginalized groups (Walls 2011). The social justice elements of our research also include a focus on promoting student access regardless of financial means and ensuring students are trained in critical community-engaged research approaches. Finally, the drafting of this article presents an approach that relies on students as curriculum designers, authors, and co-researchers. Similar to research completed in local communities, students are not frequently part of the higher education research agenda process even though the research is designed to serve them.

The implications of the model for our own higher education institutions include the development of additional "Conference as Curriculum" cohort events that feature funding to incentivize and support student attendance. Faculty and staff should consider this conference-based model as a way to deepen student learning both as academics and practitioners. This conference experience presented a rich source of networks/contacts, knowledge, and can serve as a template for future initiatives that may utilize experiential learning to shape the next generation of community-engaged practitioners and researchers. Alternatively, other practitioners and even conference planners might also learn from this experience in shaping conferences as sites of learning.

## 6. Conclusions

This article explores the use of a "Conference as Curriculum" model (Campbell et al. 2021) to develop student understanding of challenges and opportunities in the field of community-engaged research at two higher education institutions in Monterey County, California. CoLab supported 22 students to participate in the All-In Conference in October 2022. This conference provided the setting for graduate and undergraduate student scholars to be engaged with and learn about the frameworks and practices of critical community-engaged research and community–university partnerships.

Our student-author vignettes provide insight and potential lessons learned from this intervention, emphasizing the importance of building a peer cohort structure and dedicating institutional funding to incentivize and ensure equitable access for students to such experiential learning activities. The student perspectives highlight a deepened understanding of CER, the fostering of connections within their cohort and with other conference participants, and the development of professional acumen. Such insight aligns with existing research positing conferences as sites of learning and socialization of scholars (Campbell et al. 2021; Chapman et al. 2009; Cherrstrom 2012; Kuzhabekova and Temerbayeva 2018; Omolabake et al. 2019) and provide emerging support for the use of experiential learning interventions for CER training through student attendance at conferences. Future research should be conducted to provide qualitative and quantitative empirical support to the "Conference as Curriculum" model (Campbell et al. 2021), especially in regard to the model leading to students developing a deeper understanding of CER and the implications for higher education institutions, scholars, and conference planners.

Finally, this paper describes CoLab's six-year journey as a joint initiative at two universities in the same California county and their work on an emerging model that supports Gordon da Cruz (2017)'s model for critical community-engaged research practices amongst community and community partners and higher education students, faculty, and staff. CoLab experienced "choques" (or learning debates) during the critical participatory action research process as described by Fine and Torre (2019). We worked to use a critical participatory action research approach and ethos during the design process and through CoLab's multiple iterations and community projects. The article describes how

the community-engaged research initiative sought to address the administrative burden that community-engaged student projects place on local community partners, adding to the evolving academic discourse surrounding challenges in community-university partnerships and institutionalizing critical community-engaged research.

**Supplementary Materials:** The following supporting information can be downloaded at: https://www.mdpi.com/article/10.3390/socsci12060352/s1, Figure S1: Goal-Setting Framework Template. The template consists of a table with the following sections: "Goals," "Research Questions," "Areas of Interest," and various "Break Out Sessions," which can be detailed in three categories including "Priority," "Alternative," and "Connection(s) to Goal." Lastly, the framework provides a space to identify present interests. This document was shared with students as a pre-conference planning tool. Students set goals and areas of interest, then reviewed the schedule of speakers and connected the speakers with their research interests to select which panels they would attend and who the students would like to meet during the conference.

**Author Contributions:** Conceptualization, M.Z. and C.T.M.; Writing—Original Draft Preparation, M.Z., C.T.M. and J.H.-W.; Writing—Review & Editing, M.Z., C.T.M. and J.H.-W.; Visualization, M.Z.; Project Administration, M.Z. and C.T.M. All authors have read and agreed to the published version of the manuscript.

**Funding:** This research received no external funding.

**Institutional Review Board Statement:** The Institutional Review Board at Middlebury College determined that this research did not include human subject research as defined by our IRB.

**Informed Consent Statement:** Not applicable.

**Data Availability Statement:** Not applicable.

**Acknowledgments:** The experiential learning activity described in this paper of practice was made possible by funding from an institutional grant from the Provost's Office at Middlebury. This grant also funded the work hours logged by the graduate assistant and primary author of the paper. We also acknowledge the support of the service-learning center and undergraduate research initiatives at California State University Monterey Bay. The authors would also like to acknowledge the feedback and support of the rest of CoLab's planning team members.

**Conflicts of Interest:** The authors declare no conflict of interest.

## Note

1     CoLab Planning Grant. 2016. Submitted to Community Foundation of Monterey County by faculty and staff at the Middlebury Institute of International Studies and Cal-State Monterey Bay.

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
