# Peer review of "Experiential Learning: Conferences as a Tool to Develop Students’ Understanding of Community-Engaged Research"

_socsci, doi:10.3390/socsci12060352_

Round 1

Reviewer 1 Report

Thank you for the creative and interesting work. Please see the following comments for your consideration:

Title: Research title is good and appealing

Abstract: In the abstract, it will be good to have the methods used in conducting the study embedded before sharing information about the findings. Recommendations/suggestions for improving practices through the results will add value to the abstract section.

Introduction: Are there gaps in the literature necessitating the study? Adding more justifications for the study will be beneficial.

Intervention and Implementation: Intervention? Are there issues necessitating the intervention? Adding that information in the introduction could have been considered. Intervention design consideration and the remaining sub-sections could have mainly focused on what the labels depict. Primary data generated through students’ reflections could have been presented in the lessons learners as the results section. 

In addressing these concerns, it will be essential to discuss the data presented was collected and analyzed briefly. The decisions made should also be justified using the extant literature.

Lessons learned from the intervention: This section will be enhanced when the data collection and analysis strategies are clarified and supported with other scholarly works. Discussing the results along with the extant literature will also add value to the article.

Other sections read well: The limitations related to methods could have been addressed by clarifying the methods used in collecting and analyzing the data reported in the lessons learned section. Ethical factors considered also needs clarification and could possibly be highlighted in the methods section.

Author Response

Thank you for reviewing our article. The authors of this article have removed reference to a student conference satisfaction survey and the associated thematic analysis. The paper now fits more squarely as a paper of practice or “project report.” The removal of the survey data should address feedback regarding the research design, questions, and methods and some of the issues related to the discussion of findings. The paper presents an experiential learning model and the observations of two of our three authors who attended the conference as graduate-student participants. 

Abstract: After detailed discussions, the authors are confident that the paper is better without the inclusion of the student survey results. Given the more narrow focus of our paper, we are no longer presenting empirical survey findings. We share an experiential learning activity and its impact on two of our authors who participated in the experiential learning/conference opportunity. We made this approach more clear in the abstract. 

Introduction:  In response to a request to more explicitly outline the gaps in the literature, we show a gap in the literature surrounding the use of conferences as structured learning experiences for undergraduate and graduate students outside of PhD programs. We mention that the justification for the study is a gap in experiential learning programming (which typically focuses on internships, project-based learning, service-learning, and client projects) around professional conferences. We show a goal of experiential learning programming as moving students through the Zone of Proximal Development (Vygotsky 1978) to connect more with their professional field. Conferences provide the opportunity for students to engage with other professionals, learn about key discussions in their sector, and emerging topics in the field. The act of attending a conference can build a student’s confidence in joining a particular field. The converse might also be true, as a student may not find the field of interest based on the conference content and the norms of values of a particular sector. The organizers of the experiential learning activity perceived a need for students to attend a professional conference (many were not able to do so in person due to COVID-19), learn about community-engaged research practices from practitioners, and gain knowledge on the social justice sector and the types of jobs and work in this field. 

We added this language to the introduction to better present the paper. 

We also discuss how the initiative planning team explored approaches to community-engaged research (CER) at an institutional level including applying a critical lens throughout the Initiative design and implementation process. The article uses a naturalistic and participatory action research approach to present findings from two of our authors on their experience as student participants in the conference as curriculum experiential learning activity. Our review of extant literature and the inclusion of observations from student participants in the experiential learning activity presents one model for how higher education institutions and experiential learning practitioners can build learning experiences around participation in professional conferences. This experiential learning model also shows how a conference as curriculum activity in the social justice space can expose students to critical community engaged research processes and career paths. 

Intervention and Implementation: We recognize that the word “intervention” has a different meaning in Psychology and the Social Work sectors than the word does for project managers or in the organizational operations space. For this reason, we will use “experiential learning activity” in place of “intervention” for clarity. We included more references to extant literature. 

Lessons learned from the activity: We removed the thematic analysis to streamline the project report. We do include reflections from two of our authors who participated in the conference. We have edited the article to include references to extant literature alongside our results whenever possible. 

Other Sections Read Well: We removed survey data, thereby addressing the methods questions. We clarified the ethical factors section so that this section directly addresses any concerns. 

Author Response

Thank you for your thoughtful comments. 

  1. The purpose of the paper of practice is to share the conference as curriculum format and to share the initiative’s larger community-engaged research model. We do not evaluate the initiative’s model in this paper, merely present it to provide framing for the experiential learning activity. You mention the goals of the initiative as also goals of the experiential learning activity, but the mentioned goals refer to the longer term goals of the cross-institution community-engaged research initiative. The experiential learning activity does attempt to address some of these goals, but by no means addresses all of the goals. We have made this more clear in the introduction and conclusion to avoid confusion. We also share the experiential learning model finances in more detail and couch the cost among other higher education expenses. By presenting the overall cost of the event, the intervention may not seem as daunting from a financial perspective given the other budget items in a university budget that greatly surpass this expense but may or may not show impact. We also reference information on conference funding within US higher education institutions and point to the regional benefit. We also show how student worker time can be used to benefit the student and keep costs down. We expand on the model to describe potential approaches to serve larger groups of students. Ultimately it is up to universities to decide whether their budget should promote this kind of activity. We recommend universities take stock of other experiential learning activities and budget items to trace their impact on the student experience and learning outcomes. We present a relatively cost-effective model that served 22 students. A breakdown of fees charged to students might show an opportunity to build in the cost of experiential learning into the cost of attendance or rethink higher education models to employ more outside learning opportunities that can be effectively deployed without major universities resources. Online degree programs should explore the promotion of professional conference attendance as a means to connect students to their professional communities of practice, particularly if they are not already working in the field. 
  2. We added information on how the model could be embedded in the existing curriculum and expand upon the full cost (which we did not detail in our original copy). 
  3. We added language on the limitation of students participating on only one day. As participants ourselves, each day was a worthwhile learning experience—but yes, we acknowledge that one day vs. three days provides a different experience and potentially less learning and connection due to the shorter duration and exposure to less content. We reference the potential that one day is sufficient to accomplish the goal of exposing the student to the community of practice, approaches, and the norms, concerns, and trends of the field. As many of our participants were newcomers to the field of PAR and CAR, the ability to attend one day gave them that initial exposure to a new field that we had hoped for to plant the seed for future learning. 
  4. We were able to accommodate all students who wanted to attend and were logistically able to attend at our small graduate school. As schools emerge from the lockdown of the COVID-19 pandemic and professional conferences reemerge as spaces for connection, learning, and collaboration—we believe the use of professional conferences as part of HIP practices will grow and can be organized within degree programs similar to practicum internship requirements are within many undergraduate and graduate degrees. Schools should thoughtfully consider what touchpoints with the professional world should be integrated into a learning community. Your comments bring to light the assumptions any reader might make about the cost. We did not hire a new staff member to implement this activity. We employed existing experiential learning staff, faculty who teach in this space, and a graduate student leader to organize select activities before, during, and after the conference. The intervention was modest in staff time and financial commitment. We show what just a few hours can do to building a cohort model approach. We appreciate your feedback on including the activity within a course. We would recommend including the activity as part of a course when possible, however, we also recommend further exploration of the use of this activity as an extra-curricular learning experience. 

Reviewer 3 Report

Thank you for the opportunity to review this manuscript. Overall, this is a well-written manuscript and replicable for other institutions of higher education who want to leverage Conference as Curriculum model. This is a good framework for students attending a conference. However, I found the title to be misleading as there was ultimately little about social justice in this manuscript. 

Additional elements if addressed, could greatly improve the relevance of the manuscript:

Abstract

-          Line 18 “high education” should be “higher education” I believe

-          The abstract does not include any actual findings/outcomes data and would benefit from this addition

Introduction

The introduction is very brief, and there is no literature review. As a result, this manuscript is not well situated within the current literature on Conference as Curriculum or Participatory Action Research. While the title is appealing, the reader is not provided with any background knowledge on social justice initiatives and the link to experiential learning or university-community partnerships.

For example, line 54-55 – about service learning projects ending when classes end or students graduate. This seems like an important social justice problem but no background literature is provided. How will this model you present alleviate this problem? 

There is much more time and space dedicated to the Story of the Initiative and "Our approach" that becomes repetitive  throughout the manuscript.

The use of first person narrative throughout the manuscript is distracting: “we” and “our.” Is this traditionally part of PAR?

Interventions and Implementation

Line 169 “#5” is missing

The quote on page 6 is out of place. In general, the quotes throughout the manuscript are concerning because only the student authors’ quotes have been selected from the data set, which can be misleading and a source of bias. Why weren’t other student quotes selected? Was there IRB approval or a consent process used to share the student survey data or quotes. That was not evident in the manuscript. 

Figure 2 could be changed to an Appendix and is not needed in the body of the manuscript. The readers would likely be better served with a Table of the student survey results if there is IRB approval.

Line 353-354

This is the major area of concern for me. How was the qualitative data analyzed? What were the strategies employed to ensure trustworthiness? Were any of the authors experienced with qualitative data analysis?

Line 347-350 – belongs in the Limitation section

Discussion and Conclusion

Ultimately, there are few connections made to Social Justice. The students could have attended any conference on any topic.

Line 501-503 – this is what I was most interested to read about but ultimately is not supported by the data that precedes it.

Finally, the conclusion is too long and should be edited to avoid redundancy.

Author Response

Thank you for your feedback. We included a more explicit reference to the social justice elements of our research in the “Implications” section. Social justice is a goal of our research through the design of the university-community research initiative to the training of students on community-engaged research approaches that promote a grassroots and community-driven research agenda. We promote social justice through the promotion of student attendance at a professional conference regardless of their financial situation. The conference was also a way to train students on social justice approaches to research and the issues faced by organizations working on social justice issues. Finally, the drafting of the article presents an approach that relies on students as authors and researchers exploring the learning models often only explored by faculty and staff with students secondary in the design process. 

We included additional references on service learning and community-engaged research to expand the literature review.

We made a clearer distinction on the paper format. The article is primarily a paper of practice. We shake a community-engaged research model between two higher education institutions in the same region. We removed the student satisfaction survey and include only student author vignettes. Additional empirical research is needed outside of this study. 

We removed any repetitive information on the university community-engagement model (The “Initiative”). 

“We” and “our” can be used in academic research. This is a style preference. 

We corrected all typos. There is no #5. This was a formatting error. 

No table of survey results was included due to the removal of student survey data from the article. 

Figure 2 was moved to the appendix. 

The survey data was removed from the article, making the data analysis section a moot point. The article uses a naturalistic and exploratory qualitative approach. Co-author vignettes are included. 

We elaborated on the social justice connections throughout the article. We do caution that the operational aspects of such an initiative are not as inspiring to read about, but equally important in addressing social justice issues. 

We shortened the conclusion to remove redundancy. 

Thank you for your comments. 

Reviewer 4 Report

Experiential Learning for Social Justice

Line 169 extra number

Line 241 you have both ‘a’ and ‘the’

Line 242 is off somehow

Lines 249 and 255 appear to be saying the same things

I do not think that Figure 2 is integral to the paper I suggest deleting that figure

Heading 3/5 needs formatting

English is fine

Author Response

Line 169 extra number (Authors: Thank you, this extra number was a formatting issue and was removed.)

Line 241 you have both ‘a’ and ‘the’ (We have removed the extra word). 

Line 242 is off somehow (We cleaned up the language.) 

Lines 249 and 255 appear to be saying the same things (We deleted duplicate language.) 

I do not think that Figure 2 is integral to the paper I suggest deleting that figure

(We added Figure 2 to the appendix.) 

Heading 3/5 needs formatting

(We have reformatted the heading to be uniform. ) 

Round 2

Reviewer 1 Report

Thank you for your efforts. Please see the following comments for your consideration:

Title: experiential learning for social justice? Why the concept of social justice is added to the title is not explicit from the manuscript. You consider revising it as follows:

Conferences as an experiential learning tool to …

Abstract - should show the study's purpose, methods, findings and recommendations/implications

Introduction - While experiential learning and high impact practices have grown in their use in higher education institutions in the U.S., there remains limited literature on the use of professional conferences as structured experiential learning activities or high impact practices, particularly within professional graduate degree programs. So, the study is about the use of conference as an experiential learning tool and what social justice adds to the title is not clear. Any evidence to support the claim that there is a limited research on the subject? Adding that will also enhance the paper.

The Initiative - Through a critical self-assessment and conversations with our community partners in the years leading up to the development of the Initiative, both institutions realized the limitations of our approach. Community partners expressed concern over the lack of continuity between projects when a class ended or a student graduated. – How is this concern about equity leading to the next sentence and argument made in the manuscript? The concern seems to show the need for an ongoing support for projects beyond students’ university career. You may want to check this again.

The addition of community-based research approaches in service learning and other experiential learning programming may offer a pathway to promote social change (Jason & Glenwick, 2016). In what sense? It is not clear why this sentence was added. You may want check again.

The remaining portions of the paper could be re-arranged as follows:

The initiative and our purpose can be combined as they both focus on the research context.

Methodology section should be provided showing research methods, data collection and analysis procedures, and ethical factors considered, at least.

Results and Discussion - Showing the themes emerging from the qualitative analysis and their discussion along with the extant literature

Author Response

Thank you for your comments. We edited the title of the article to remove the words "Social Justice." We also added additional justification for the article in the introduction and removed the CER source you questioned (thank you!). 

We rearranged the content as you suggested.

We added a methodology section and described the exploratory qualitative approach employed by the student co-authors.

Reviewer 3 Report

The authors have addressed concerns. 

Author Response

In our version 3, we have added additional information to the introduction. We also described our methodology for the inclusion of student-author vignettes. The conclusion is also more robust.